# Deep Learning-Based Signal Detection for Underwater Acoustic OTFS Communication

Yuzhi Zhang [1,2,*] , Shumin Zhang [1,2], Bin Wang [1,2], Yang Liu [1,2], Weigang Bai [3] and Xiaohong Shen [4]

1    School of Communication and Information Engineering, Xi'an University of Science and Technology, Xi'an 710054, China
2    Xi'an Key Laboratory of Network Convergence Communication, Xi'an 710072, China
3    State Key Laboratory of Integrated Service Networks, Xidian University, Xi'an 710071, China
4    School of Marine Science and Technology, Northwestern Polytechnical University, Xi'an 710072, China
*    Correspondence: yuzhizhang@xust.edu.cn

**Abstract:** Orthogonal time frequency space (OTFS) is a novel two-dimensional (2D) modulation technique that provides reliable communications over time- and frequency-selective channels. In underwater acoustic (UWA) channel, the multi-path delay and Doppler shift are several magnitudes larger than wireless radio communication, which will cause severe time- and frequency-selective fading. The receiver has to recover the distorted OTFS signal with inter-symbol interference (ISI) and inter-carrier interference (ICI). The conventional UWA OTFS receivers perform channel estimation explicitly and equalization to detect transmitted symbols, which requires prior knowledge of the system. This paper proposes a deep learning-based signal detection method for UWA OTFS communication, in which the deep neural network can recover the received symbols after sufficient training. In particular, it cascades a convolutional neural network (CNN) with skip connections (SC) and a bidirectional long short-term memory (BiLSTM) network to perform signal recovery. The proposed method extracts feature information from received OTFS signal sequences and trains the neural network for signal detection. The numerical results demonstrate that the SC-CNN-BiLSTM-based OTFS detection method performs with a lower bit error rate (BER) than the 2D-CNN, FC-DNN, and conventional signal detection methods.

**Keywords:** OTFS; underwater acoustic communication; deep neural networks; signal detection; delay-Doppler domain

## 1. Introduction

The underwater acoustic (UWA) channel is one of the most challenging communication media [1,2]. The low propagation speed of UWA waves will cause the multi-path and Doppler effects to be several magnitudes larger than wireless radio communication. Even when the transceivers do not move, the seawater movement and sea surface fluctuations will still cause Doppler shift. The severe multi-path and Doppler effects will cause time- and frequency-selective fading. Since the available carrier frequencies for medium-range UWA communication are only in the kHz range, a slight movement of the transceiver will cause a large Doppler shift. Orthogonal frequency division multiplexing (OFDM) is widely applied in UWA communication due to its high spectrum efficiency and robustness against the multi-path effect [3–8], whereas for classical OFDM communication, severe Doppler shift in the UWA channel will lead to inter-carrier interference (ICI), and the performance of OFDM will degrade significantly.

Orthogonal time frequency space (OTFS) modulation is a promising two-dimensional (2D) modulation technique proposed in recent years for high-mobility communication scenarios [9,10]. The basic principle of OTFS is to modulate information symbols in the 2D delay-Doppler (DD) domain rather than the time frequency (TF) domain. In light of the DD domain, OTFS modulation can transform the channel into an approximately

non-fading channel through a series of 2D transformations. In UWA OTFS communication, the fast time-variant UWA channel will still bring ICI and inter-symbol interference (ISI). To improve the communication performance of OTFS, channel equalization and signal detection can mitigate the interference. Signal detection algorithms generally include linear and nonlinear detection algorithms. Linear signal detection methods, such as the zero-forcing (ZF) algorithm [11] and linear minimum mean squared error (LMMSE) [12], have high complexity in practical implementation. Bayesian-based nonlinear algorithms assume the interference terms are approximately Gaussian distributed noise, such as message passing (MP) [13] and the Markov chain Monte Carlo algorithm. However, in actual UWA communication systems, the interference term may not obey the Gaussian distribution. Although the nonlinear algorithms can approximate the optimal performance with a large number of iterations, the complexity is much higher than that of the LMMSE algorithm. In UWA OTFS communication, signal detection has been studied by linear equalizers [14–16] under different UWA channels.

Machine learning can be used in wireless communication for signal detection. In [17], supervised machine learning techniques were applied to decode the tag symbols. The input features that form the training data were explored and extracted from the received signal for machine learning-based detectors. In [18], support vector machine (SVM)-based data detection is proposed for optical OFDM in visible light communication. In this paper, the SVM detector contained multiple binary classifiers with different classification strategies. The experiment results presented that the SVM detection offered improved BER performance compared with the traditional direct decision method. In [19], the SVM in machine learning was used to jointly optimize the processing chain of signal detection, feature extraction and signal classification, and the simulation results show that the SVM had good performance. However, when the sample size is large, machine learning has difficulty dealing with the problems, and deep learning (DL)-based methods can solve such problems well. For example, the convolutional neural network (CNN) and its derivative algorithms can automatically learn the deep features of input digital information for subsequent classification [20]. The recurrent neural network (RNN) is also widely used due to its advantages in processing time series data [20].

In recent years, the DL-based method has shown its potential in communication systems [21]. Ye et al. replaced the channel equalization and demodulation blocks of the receiver with a five-layer fully connected deep neural network (FC-DNN) in the OFDM system [22]. The experiment results show that the DNN-based receiver was more reliable than the conventional methods. In the UWA OFDM communication system, an FC-DNN is used to realize the whole signal processing at the receiver [23,24], and simulation results show that the FC-DNN offered better bit error rate (BER) performance than conventional algorithms. In [25], the long short-term memory (LSTM) neural network architecture was employed as the receiving module of the cyclic shift keying spread spectrum UWA communication system. The neural network is fed the communication signals passing through known channel impulse responses in the offline stage and then used to demodulate the received signal in the online stage. For UWA communication, the receiver in [26] jointly employed a CNN for channel equalization and an FC-DNN for demodulation. Compared with a single DNN-based OFDM receiver, this joint network model can better extract channel information for data recovery. In [27], DenseNet was proposed to replace the entire information recovery process of block-based MIMO receivers. DeseNet takes multiple modules as one system for joint optimization, and its BER outperforms block-by-block receivers. A signal detection scheme based on LSTM was proposed in [28]. The authors utilized an RNN with the BiLSTM architecture for signal detection in [29]. The simulation results show that the trained model can trace the characteristics of wireless time-varying channels and achieve accurate and robust signal recovery performance.

For using DL in OTFS systems, Naikoti et al. conducted a preliminary exploration of FC-DNN-based signal detection [30]. Li et al. proposed a receiver with CNN-based signal detection for OTFS [31]. Y. K. Enku et al. proposed a 2D-CNN-based OTFS signal detection

scheme [32] and utilized data augmentation to improve the overall performance. In [33], an FC-DNN was used to replace the signal detection in the UWA OTFS system. It can be seen that the DL-based methods outperformed the conventional methods under complex channels. Whereas the DNN and CNN can only extract local features, this paper proposes an OTFS signal detection scheme based on the joint CNN and RNN to utilize both local and sequential features.

The main contributions of this paper are summarized as follows:

- We propose an UWA OTFS signal detection method based on the deep neural network. The UWA channel has severe transmission loss, time-varying multi-path propagation and the severe Doppler effect, which are extremely challenging for signal detection. Conventional signal detection methods not only have high computational complexity but also require prior knowledge of the noise. DL-based signal detection has the advantage of recovering signals with complex nonlinear interference and noise by training and learning, and it does not have to assume any prior knowledge. In this paper, we propose a DL-based signal detection method for UWA OTFS in the complex, nonlinear UWA channel to improve system performance. To the best of our knowledge, this is the first DL-based signal detector proposed for UWA OTFS communication.
- The SC-CNN-BiLSTM network is designed for OTFS signal detection in the complex UWA channel, which takes the advantages of both CNNs and RNNs for feature extraction and sequential data processing. Different from our previous work [33], a totally new neural network structure is proposed for performance improvement. The CNN in the proposed network can extract data features and learn the potential relationship between its input and output. Furthermore, the skip connection (SC) in a CNN can provide the flexibility of data feature fusion for performance improvement. The cascaded BiLSTM in the network can memorize and extract the effective information from sequential transmitted symbols from the past to the future, which can mitigate the ICI and ISI. For UWA OTFS communication, SC-CNN-BiLSTM signal detection outperforms other previous proposed DL-based and conventional linear and nonlinear signal detection methods.

The remainder of this paper is organized as follows. Section 2 presents the UWA OTFS system model. Section 3 proposes the SC-CNN-BiLSTM scheme for signal detection. Section 4 evaluates the performance of SC-CNN-BiLSTM-based signal detection with the simulation and experimental data and compares its performance with other signal detection methods. Section 5 provides a discussion about the results. Section 6 concludes our research.

## 2. UWA-OTFS System Model

Compared with wireless radio communication, the Doppler shift in UWA communication is more severe, which is mainly determined by the transmission characteristics of the UWA channel [33]. Table 1 shows the characteristics comparison between wireless radio and the UWA channel. The propagation speed of UWA waves is five orders of magnitude slower than that of radio waves. Due to the severe distance- and frequency-dependent attenuation, the available frequency for long-range communication is only in roughly the KHz range. Due to these factors, even a slight movement can cause obvious Doppler shifts in the UWA channel.

**Table 1.** Comparision of radio and UWA communications.

| Parameters | Wireless Radio Communication | UWA Communication |
|---|---|---|
| Propagation speed $c$ | $3 \times 10^8$ m/s | 1500 m/s |
| Typical carrier frequency $f_c$ | Several GHz | Several kHz |
| Subcarrier spacing $\Delta f$ | Several KHz | Several Hz |
| CFO $f_D$ with $v_t = 1$ m/s | Several Hz | Several Hz |
| Normalized CFO $\theta$ with $v_t = 1$ m/s | $\approx 1 \times 10^{-3}$ | $\approx 1$ |

In Table 1, the carrier frequency offset can be calculated as $f_D = (v_t f_c)/c$, where $v_t$ is the speed of movement between the transceivers. The normalized CFO can be calculated as $\theta = f_D/\Delta f$, which can represent the impact of the CFO referenced to subcarrier spacing. For example, as shown in Table 1, for a relative moving speed of 1 m/s, the normalized CFO is about $10^{-3}$ for radio communication and 1 for UWA communication. Doppler shifts on subcarriers have more severe impact on UWA communication performance than radio systems. In the UWA OTFS system, the Doppler effect will cause severe ISI and ICI. This paper will enhance the performance of the OTFS system with the power of deep learning.

Figure 1 shows the block diagram of the UWA OTFS system. At the transmitter, the modulation module can map the one-dimensional constellation symbols $x = [x_1, ..., x_{NM}]$ into 2D transmission symbols $x[k, l]$, $k = 0, ..., N - 1, l = 0, ..., M - 1$ with the specified modulation mode (e.g., BPSK or QPSK). The 2D symbols are distributed over $N \times M$ OTFS delay-Doppler data grids. Then, the symbols in the DD domain are converted to the TF domain by an inverse symplectic finite Fourier transform (ISFFT) as

$$X[n, m] = \frac{1}{\sqrt{MN}} \sum_{k=0}^{N-1} \sum_{l=0}^{M-1} x[k, l] e^{j2\pi(\frac{nk}{N} - \frac{ml}{M})}. \tag{1}$$

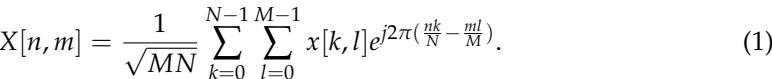

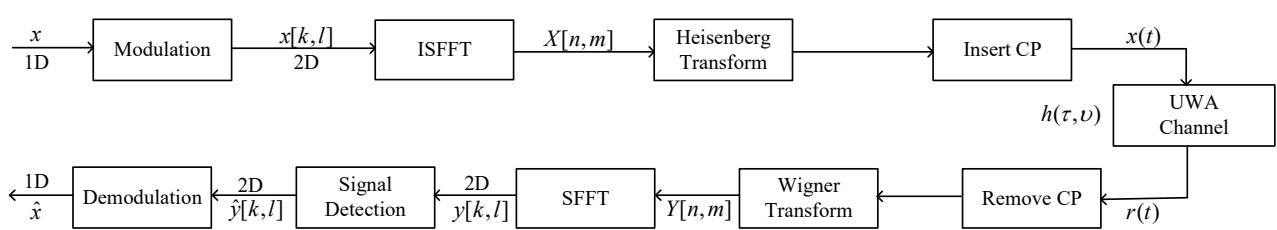

**Figure 1.** Block diagram of the UWA OTFS system.

The TF domain signal is further transformed into time domain signal $x(t)$ by Heisenberg transform as

$$x(t) = \sum_{n=0}^{N-1} \sum_{m=0}^{M-1} X[n, m] g_{tx}(t - nT) e^{j2\pi m\Delta f(t - \Delta f)}, \tag{2}$$

where $\Delta f$ is the subcarrier spacing, $T = 1/\Delta f$ is the symbol duration and $g_{tx}$ is the transmit pulse-shaping filter.

The channel impulse response (CIR) $h(\tau, v)$ in the DD domain can be expressed as

$$h(\tau, v) = \sum_{i=1}^{p} h_i \delta(\tau - \tau_i) \delta(v - v_i), \tag{3}$$

where $h_i$ is the channel coefficient of path $i$ and $v_i$ and $\tau_i$ are the frequency bias and time delay of path $i$, respectively. We assume that the CIRs are perfectly known at the receiver.

The transmitted signal will go through the UWA channel, which is represented by the CIR and additive noise. The received signal can be expressed as

$$r(t) = \sum_{i=1}^{p} h_i e^{j2\pi v_i(t - \tau_i)} x(t - \tau_i) + w(t), \tag{4}$$

where $w(t)$ is the additive Gaussian white noise.

At the receiver, the received time domain signal $r(t)$ is converted into a TF domain signal through a Wigner transform as

$$Y(n, m) = \left[ \int g_{rx}^*(t - \tau) y(t) e^{-j2\pi f(t - \tau)} dt \right], \tag{5}$$

where $\tau = nT$, $v = m\Delta f$.

Then, the TF domain signal is converted into a DD domain signal by a symplectic finite Fourier transform (SFFT) as

$$y[k,l] = \frac{1}{\sqrt{MN}} \sum_{n=0}^{N-1} \sum_{m=0}^{M-1} Y[n,m] e^{-j2\pi\left(\frac{nk}{N} - \frac{ml}{M}\right)}. \tag{6}$$

After parallel-to-serial conversion, $y[k,l]$ is converted to $y$ at a size of $N \times M$.

Finally, signal detection and demodulation will be performed to recover the transmitted signal as $\hat{x}$.

The multi-path effect and Doppler shift in UWA communication are more severe than that of radio communication. In UWA communication, assume that the maximum delay of the time-varying UWA channel is $\tau_{max}$ and the maximum Doppler shift is $v_{max}$. In OTFS modulation, from the view of the DD domain, the OTFS parameter design is related to the channel conditions. In the Doppler axis, $1/T$ determines the maximum supportable Doppler shift as $v_{max} < 1/T$. In the delay axis, $1/\Delta f$ determines the maximum supportable multi-path delay as $\tau_{max} < 1/\Delta f$. In time-varying UWA channels, the maximum multi-path delay $\tau_{max}$ is large, so the corresponding designed value of $\Delta f$ should be small, and T is as large as $T = 1/\Delta f$.

To support a certain data rate of $NM$ subcarriers per frame, the OTFS system is designed with a total bandwidth $B = M\Delta f$ and frame duration $T_f = NT$. $\Delta f$ is small, so the setting of $M$ should be large for high data rate communication. $T$ is large, and the frame duration $T_f$ should not be too large for demodulation latency, so $N$ cannot be too large. For effective OTFS communication in the UWA channel, a small value of $N$ and large value of $M$ should be selected to achieve efficient communication.

Based on the above analysis of UWA OTFS, a large $M$ and small $\Delta f$ result in a high resolution for the frequency, which is sensitive to intercarrier interference. Additionally, when $T$ is large and the value of $N$ is small, the resolution of the corresponding Doppler shift decreases, which will affect the accuracy of signal detection. For the challenging UWA OTFS communication, this paper designs a DL-based signal detector for data recovery in the UWA channel with severe interferences.

## 3. SC-CNN-BiLSTM-Based Signal Detection for UWA-OTFS

Figure 2 shows the proposed deep learning-based OTFS system, where the transmitter is the same as the typical OTFS system and the detection module is replaced by SC-CNN-BiLSTM. We assume that the CIR is known in the detection module.

SC-CNN-BiLSTM training is performed using a set of training sets known at the transmitter and receiver. The training data are pseudo-randomly generated by the transmitter and sent to the receiver through the DD channel. The received signal vector $y$ and the transmitted signal vector $x$ can be used for training the neural network. After being trained, the real and imaginary parts of $y$ in the validation set are used as input to the SC-CNN-BiLSTM to recover the unknown transmitted data.

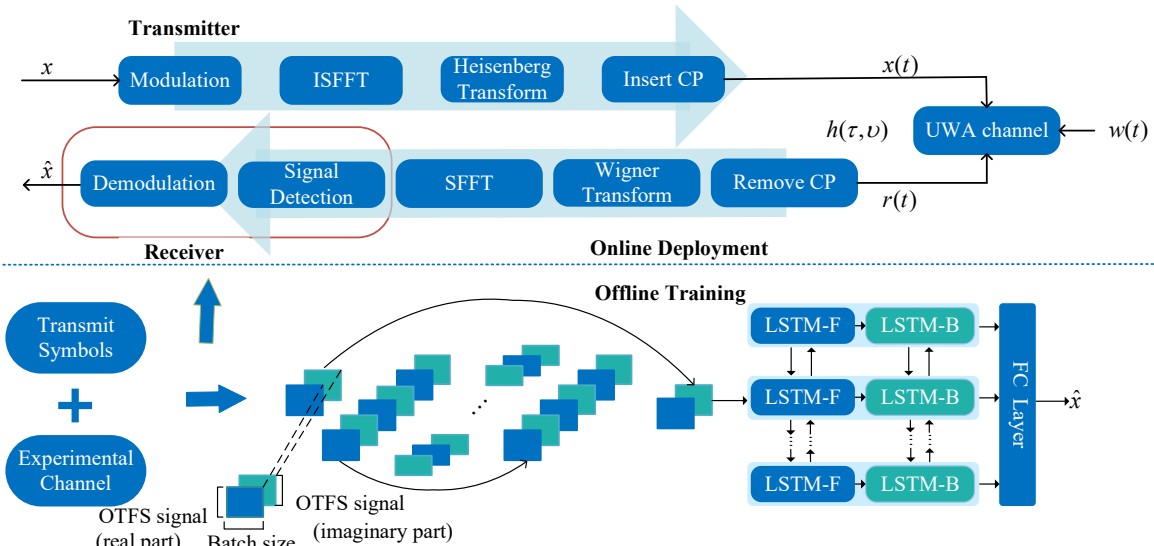

**Figure 2.** Deep learning-aided UWA OTFS communication.

### 3.1. Architecture of the Proposed SC-CNN-BiLSTM Detector

For UWA OTFS with severe Doppler effect, a DL-based channel detection method is designed by cascading the skip connection CNN and BiLSTM. The architecture of SC-CNN-BiLSTM for UWA OTFS is shown in Figure 2. It includes the following layers:

- **SC-CNN layer**: The SC-CNN layer extracts local signal features and learns the hidden relationship between the input and output.
- **BiLSTM layer**: The BiLSTM layer can extract features of time series data from both the forward and backward directions and keep correlated and ignore uncorrelated information by the gates structure. It can mitigate interference for UWA OTFS.
- **FC layer**: A fully connected layer with a sigmoid activation function is used to output soft bits for signal detection.

**SC-CNN layer:**

The CNN is a type of feedforward neural network structure with convolution calculations. With the advantage of convolution operation, the CNN can extract and express the internal complex correlation of signals, which plays the role of the mapping function. Meanwhile, its weight-sharing structure significantly reduces the number of weights and network complexity. In the SC-CNN layer of the proposed SC-CNN-BiLSTM network, the CNN consists of three convolutional neural network layers and three deconvolutional neural network (DeCNN) layers. The multiple convolutional layers are used to extract the signal features and internal correlations. The hidden layers in the neural network do not output the exact value. The output of the previous layer is the input of the next hidden layer. Accordingly, the output of the CNN can be expressed as

$$\tilde{x} = f_6(f_5(...f_1(y))), \tag{7}$$

where $y$ is the input data, $\tilde{x}$ is the output of the CNN, and the function $f_n(\cdot)$ represents the operation in each convolutional layer.

In the neural network structure, SC can create short paths from previous layers to later layers. Not only can they reuse information for training, but they also can ease the gradient disappearance problem in network backpropagation. In the proposed SC-CNN layer, we add symmetrical skip connections to transfer learned feature mapping from the previous layer to the current layer. Each DeCNN not only takes the output from the previous layer as input but is also skip connected to the previous CNN layers. This mechanism enhances feature reusability. With the output in the same dimensionality from the earlier layers, the

SC-CNN can learn more effective information through the interactions of layers. As shown in Figure 2, in SC-CNN-BiLSTM, the first DeCNN layer takes the output of the previous CNN layer as its input. Starting from the second DeCNN layer, the fused feature vector of the SC-CNN can be expressed as

$$DeCNN_l = g(DeCNN_{l-1} + CNN_{n+1-l}), \tag{8}$$

where the function $g(\cdot)$ represents the feature fusion of different network layers, $n$ is the number of layers for the CNN or DeCNN and the total number of layers is $2n$, while $l$ represents the $l$th CNN or DeCNN layer. The input of $DeCNN_l$ is the fusion of output of $DeCNN_{n-1}$ and $CNN_{n+1-l}$.

**BiLSTM layer:**

As shown in Figure 3, in an SC-CNN-BiLSTM cascaded neural network, the BiLSTM layer includes two LSTM networks in different directions: LSTM-F and LSTM-B. The input sequences are passed into LSTM-F in the forward direction and LSTM-B in the backward direction. These two LSTM cells are cascaded and passed to more Bi-LSTM layers. In the forward layer, the calculation is performed from the start time to the time $t$, and the output of the forward hidden layer at each time is obtained and saved. The backward layer is calculated in reverse along the time axis, and the output of the backward hidden layer at each time is also obtained and saved. Finally, at each moment, the final output can be achieved by combining the corresponding output results of the forward layer and the backward layer, which can be expressed as

$$h_t^F = f(w_1 \tilde{x}_t + w_2 h_{t-1}) \tag{9}$$

$$h_t^B = f(w_3 \tilde{x}_t + w_4 h_{t+1}) \tag{10}$$

$$x_t^{Bi} = o_t \left( w_5 h_t^F + w_6 h_t^B \right) * \tanh(C_t) \tag{11}$$

where $h_t^F$ and $h_t^B$ represent the output of the forward calculation and backward calculation at time $t$, respectively, $x_t^{Bi}$ represents the final output of the BiLSTM, $\tilde{x}_t$ represents the input of the current LSTM, $h_{t-1}$ represents the output of the last LSTM, $h_{t+1}$ indicates the output of LSTM in opposite directions, $w_1, w_2, w_3, w_4, w_5, w_6$ are the corresponding weights of the variables, $C_t$ is the cell state in LSTM and $o_t$ is the forgetting factor. BiLSTM can learn more comprehensive intrinsic correlation of the input series signal by learning from the past to the future and from the future to the past. Therefore, it can improve the performance of signal detection in UWA OTFS.

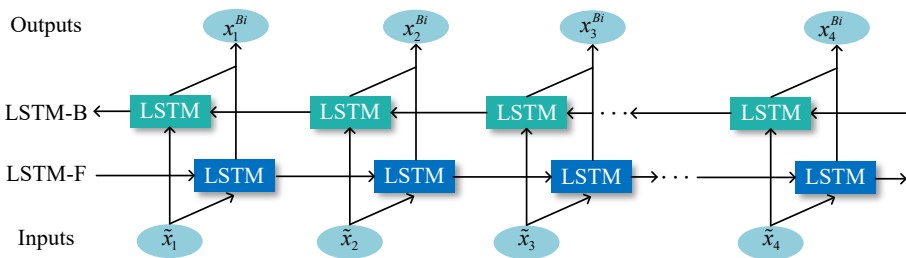

**Figure 3.** BiLSTM structure.

BiLSTM consists of mutiple LSTM cells. Figure 4 shows the basic structure of the LSTM cell. With collaboration of the input gates, forget gates and output gates, LSTM can memorize important information and solve the problem of long-term dependence on data in the learning process.

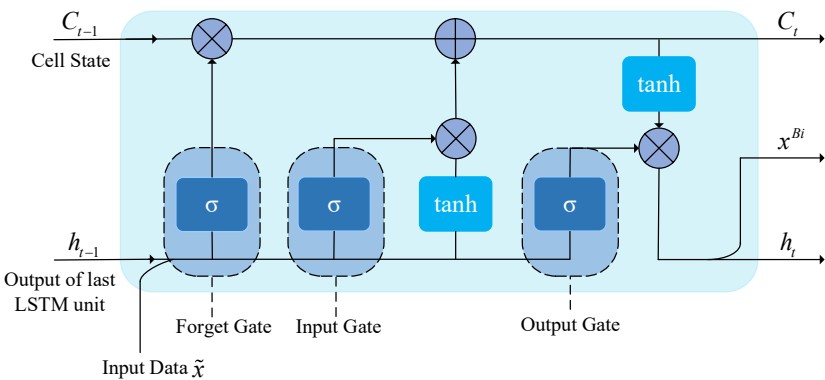

**Figure 4.** LSTM structure.

First, the forget gate of LSTM decides which information to forget to have $f_t$ for the cell state update. Then, the input gate updates the important information $i_t$ in the learning process and determines the useful information retained in cell state $\bar{C}_t$. The cell state is calculated as

$$C_t = f_t * C_{t-1} + i_t * \bar{C}_t \tag{12}$$

Finally, the output gate calculates the forgetting factor $o_t$ according to $h_{t-1}$ and $\tilde{x}_t$, and it obtains the final output $h_t$ according to $o_t$ and the cell state $C_t$:

$$o_t = \sigma(w_o \cdot (h_{t-1}, \tilde{x}_t) + b_o) \tag{13}$$

$$h_t = o_t * \tanh(C_t) \tag{14}$$

where $w_o$ is the weight of $o_t$, $b_o$ is the bias of $o_t$ and $\tanh(\cdot)$ is the hyperbolic tangent activation function.

The BiLSTM layer in SC-CNN-BiLSTM has a strong ability to capture the correlations of times series data. It can not only remember correlated and ignore uncorrelated information by the gates structure, but it also can extract features from both the forward and backward directions. Therefore, for the signal detection of sequential data with interference, BiLSTM can enhance the neural network to better memorize and extract the effective information with sequentially transmitted symbols from the past to the future, which can mitigate the ICI and ISI in UWA communication.

**FC layer:**

For the output of the SC-CNN-BiLSTM network, there is one FC-DNN layer with 32 neurons and a logistic sigmoid activation function. The 32 neurons correspond to 32 bits to be estimated from 32 consecutive subcarriers. The logistic sigmoid function mapped the output values between [0,1] as soft bits, and then the soft bits will be processed to obtain the target sent bits, which can be expressed as

$$\hat{x}' = Sigmoid\left(x^{Bi}\right) \tag{15}$$

where $x^{Bi}$ represents the output of BiLSTM.

Finally, the transmitted bits are obtained according to the decision formula as

$$\hat{x} = \begin{cases} 1, & \hat{x}' - 0.5 \geqslant 0 \\ 0, & \hat{x}' - 0.5 < 0 \end{cases} \tag{16}$$

*3.2. Training of the Proposed SC-CNN-BiLSTM Neural Network*

In the training stage, the number of training examples is chosen through trials. We start with a small number of examples and increase the number of examples until the SC-CNN-BiLSTM training tends to be stable. SC-CNN-BiLSTM learned the mapping

relationship between the received vector and the corresponding transmitted vector. After training, the SC-CNN-BiLSTM can be used for signal detection. In the testing stage, the transmitter generates random information bits, modulates the bits by OTFS and transmits the OTFS signal over the UWA channel to the receiver. The receiver utilizes the trained SC-CNN-BiLSTM to detect symbols in the DD domain for data recovery.

The performance of a neural network depends greatly on the training process. First, the loss function should be reasonably designed to provide an accurate measure of the distance between the outputs and true labels. The training process aims to minimize the difference between the original transmitted data sequence $x(b)$ and the signal detection output $\hat{x}(b)$ through the deep learning model. In this study, we define the loss function $L(loss)$ as

$$L(loss) = \frac{1}{N_B} \sum_{i=1}^{N_B} [\hat{x}_i(b) - x_i(b)]^2, \tag{17}$$

where $N_B$ is the batch size and $x_i(b)$ represents the bits in the $i$th batch.

In addition, the hyperparameters related to the network structure and training will affect the capabilities of neural networks. The learning rate affects the convergence rate and results of the DL network. The adaptive learning rate strategy is employed, which can avoid being trapped in the local optimum. In our training, the initial learning rate was set to 0.001, and the decay factor was set to 0.1.

For training optimizer selection, we compared the performance of three typical optimizers: the stochastic gradient descent (SGD) optimizer, adaptive momentum (Adam) optimizer and root mean square propagation (RMSprop) optimizer. The test was conducted with an OTFS dataset that went through an experimental channel. As shown in Figure 5, with the SGD optimizer, the loss of the proposed neural network did not converge well during training process. The convergence results of the Adam optimizer were better than those of the SGD optimizer with much lower loss, and the convergence of the RMSprop optimizer was the best. Therefore, our proposed SC-CNN-BiLSTM employed the RMSprop optimizer for training.

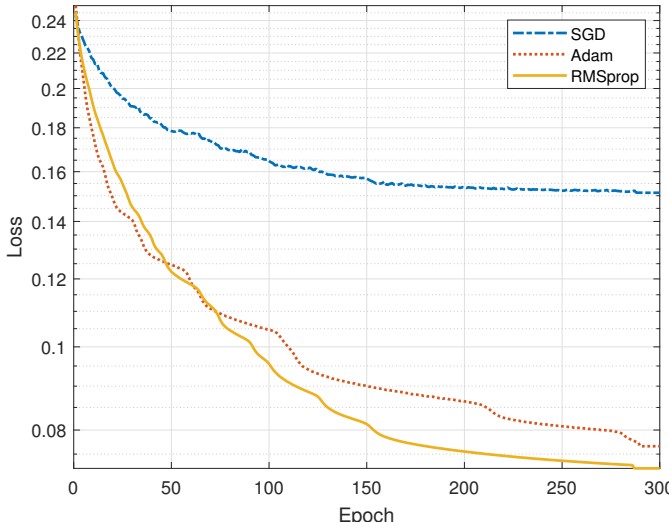

**Figure 5.** Training loss of several optimizers.

## 4. Numerical Results

### 4.1. System Set-Up

Both the simulation channel and experimental channel were used to evaluate the performance of the proposed signal detection scheme. The OTFS frame size was set to $(N, M) = (8, 64)$, which means each frame had 8 symbols and 64 subcarriers in the TF domain. The carrier frequency was set to 6 kHz. The maximum multi-path delay in the sea

experiment was about 100 ms, so the subcarrier spacing was set to $\Delta f = 1/\tau_{max} = 10$ Hz. The sound speed was set to $c = 1500$ m/s. Binary phase shift keying (BPSK) was utilized for symbol constellation mapping.

The proposed model was implemented on the DL framework of TensorFlow and Keras for training and testing. The parameters of the neural network are listed in Table 2. In the proposed SC-CNN-BiLSTM, there are $2MN$ input neurons, where $(M, N)$ is the frame size of the OTFS. For the output of the SC-CNN-BiLSTM-based OTFS detector, every 32 bits of transmitted data were grouped and predicted according to the separately trained model and then serially converted to the final output. In the proposed SC-CNN-BiLSTM, the former three convolutional layers used 4, 8 and 16 filters and the rectified linear unit (ReLU) activation function, the convolutional kernels were 4, 2 and 2, and the stride sizes were 4, 2 and 2, respectively. The latter three deconvolution layers had 8, 4 and 2 filters, the convolutional kernels were 2, 2 and 4, and the stride sizes were 2, 2 and 4, respectively. BiLSTM includes three BiLSTM layers with 30, 20, and 16 hidden units, and layer normalization (LN) was added between each BiLSTM layer to accelerate convergence and prevent overfitting. The BiLSTM layers used a hyperbolic tangent (tanh) activation function. For the output, there was one FC-DNN layer with 32 neurons and a logistic sigmoid activation function. The logistic sigmoid function mapped the output values of [0,1] as soft bits, which would be then processed to obtain the target sent bits. The output layer used the regression sigmoid function to find the predicted values of the transmitted symbols [33].

We generated 60,000 OTFS frame samples under time-varying delay-Doppler channels. The data samples were divided into the training set, validation set and test set at a ratio of 4:1:1.

**Table 2.** Parameter settings of the proposed neural network.

| Layer | Type | Input Layer | Activation |
|---|---|---|---|
| Input | reshape | - | - |
| Conv1 | Convolutional layer (4,4,4) | Input | ReLU |
| Conv2 | Convolutional layer (8,2,2) | Conv1 | ReLU |
| Conv3 | Convolutional layer (16,2,2) | Conv2 | ReLU |
| DeConv1 | Deconvolutional layer (8,2,2) | Conv3 | ReLU |
| DeConv2 | Deconvolutional layer (4,2,2) | DeConv1 + Conv2 | ReLU |
| DeConv3 | Deconvolutional layer (2,4,4) | DeConv2 + Conv1 | ReLU |
| CNNoutput | - | DeConv3 + Input | - |
| BiLSTM1 | 30 | BiLSTM1 | tanh |
| BiLSTM2 | 20 | BiLSTM2 | tanh |
| BiLSTM3 | 16 | BiLSTM3 | tanh |
| FC | 32 | FC | Sigmoid |

The BER performance of the following signal detection methods will be compared:

- Proposed SC-CNN-BiLSTM: The proposed SC-CNN-BiLSTM signal detection method for UWA OTFS;
- 2D-CNN: The DL OTFS signal detection based on the 2D-CNN proposed for wireless radio communication [32];
- FC-DNN: The FC-DNN-based signal detection method for UWA OTFS [33];
- MP: Message-passing nonlinear signal detection method for OTFS [13];
- LMMSE: Classical linear minimum mean square error (LMMSE) [12] signal detection method for OTFS;
- ZF: Classical zero-forcing (ZF) [11] linear signal detection method for OTFS.

We will evaluate the performance of SC-CNN-BiLSTM in both the simulation and experimental channels with the above system settings and also consider the non-ideal factor of signal processing in practical underwater acoustic communication.

### 4.2. Simulation Results

We considered a statistic channel simulation model in a mobile communication scenario, where the channel gains followed an independent Rayleigh distribution. The simulation's parameter settings are shown in Table 3. The maximum multi-path delay was set to $\tau_{max} = 100$ ms. There was a total number of eight random multi-paths within the maximum delay range, in which the channel gain followed an independent Rayleigh distribution. The moving speed was set to $v_m = 3.8$ knots (1 knot is 1 nautical mile per hour, which is equal to 1.852 kilometers per hour), and the corresponding maximum Doppler spread was $f_D = (v_m f_c)/c = 7.8$. The Doppler coefficient of each path was generated in $[-f_{D\,max}, f_{D\,max}]$ with equal probability.

**Table 3.** Parameters of statistic simulation channel.

| Paraments | Value |
|---|---|
| Channel gains $h_i$ | Rayleigh distribution |
| Maximum multi-path delay $\tau_{max}$ | 100 ms |
| Number of multi-paths | 8 |
| Moving speed $v_m$ | 3.8 knots |
| Maximum Doppler spread $f_D$ | 7.8 Hz |

Figure 6 shows the BER comparison of multiple signal detection methods for UWA OTFS. At the BER of $5 \times 10^{-3}$, the proposed SC-CNN-BiLSTM could achieve about 5.5 dB, 3 dB and 1.5 dB of improvement compared with the MP, FC-DNN and 2D-CNN, respectively.

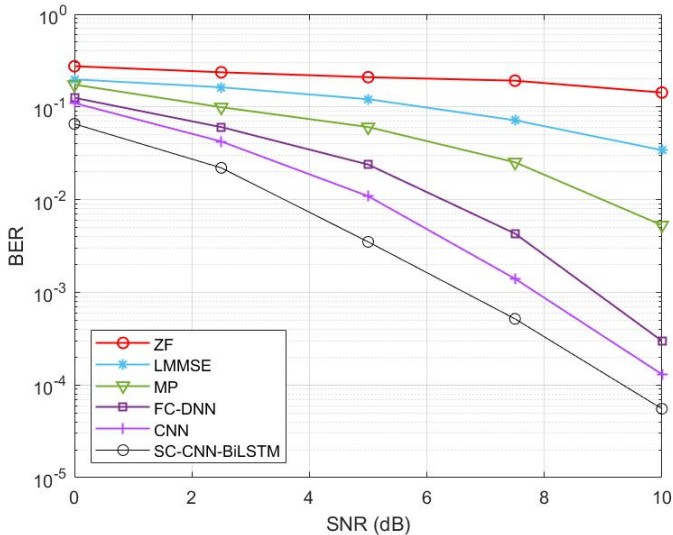

**Figure 6.** BER performance comparison for simulation channel.

All deep learning-based signal detection methods (SC-CNN-BiLSTM, 2D-CNN and FC-DNN) perform better than conventional linear ZF, LMMSE and nonlinear MP, as DL-based OTFS signal detection methods can use nonlinear operations in neural networks to better fit data in the DD domain compared with the linear-based method. Compared with nonlinear MP, DL-based OTFS detection can fit the input-output relationship through iterative optimization of the parameters and avoid falling into a local optimum for better performance.

The two CNN-based signal detection methods, SC-CNN-BiLSTM and 2D-CNN, performed better compared with FC-DNN-based signal detection. As the neurons in a CNN are connected to each other, the weights of neurons on the same feature mapping layer are shared. Therefore, the CNN can learn in parallel to avoid overfitting and achieve faster convergence. This is the major advantage of CNNs compared with other neural networks. Moreover, the CNN uses the ReLU activation function to prevent gradient disappearing.

SC-CNN-BiLSTM outperformed the 2D-CNN. A single CNN can only extract local features and cannot process time series data efficiently when used. As in SC-CNN-BiLSTM, we added symmetric skip connections to the six-layer CNN. The SC-CNN can provide more efficient information through interactions of convolutional and deconvolutional layers. After the SC-CNN extracts the important features of input vector *y*, BiLSTM can further focus on effective information in the data sequences to mitigate ICI and ISI.

### 4.3. Experimental Results

We further evaluated the performance of SC-CNN-BiLSTM-based OTFS signal detection under multiple channels from a sea experimental dataset [34]. For evaluation of the proposed scheme, we used UWA experimental channels from the WATERMARK dataset. WATERMARK is a benchmark dataset driven by at-sea measurements of the time-varying impulse response. In this paper, we employed the raw CIR measured at Norway-Oslofjord (NOF) and Kauai 1 (KAU1). The parameter settings of the experiments are shown in Table 4.

**Table 4.** Parameters of channel dataset.

| Paraments | NOF | KAU1 |
|---|---|---|
| Environment | Fjord | Shelf |
| Range | 750 m | 1080 m |
| Water depth | 10 m | 100 m |
| Transmitter deployment | Bottom | Towed |
| Receiver deployment | Bottom | Suspended |
| Doppler coverage | 7.8 Hz | 7.8 Hz |

The CIRs of the NOF channel in the time domain and DD domain are shown in Figures 7 and 8. As shown in Figure 7, the CIR of NOF in the time domain had an obvious time-varying multi-path. Figure 8 presents the corresponding CIRs in the DD domain, where the Doppler shift for each path can be observed.

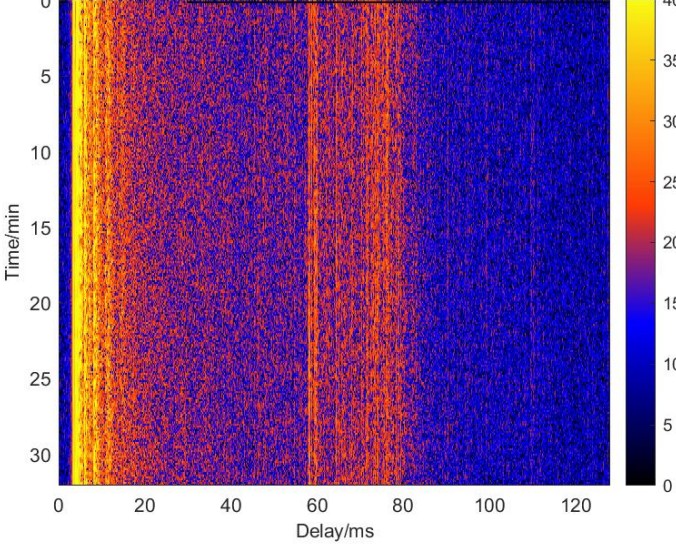

**Figure 7.** CIR in time domain under NOF experiment.

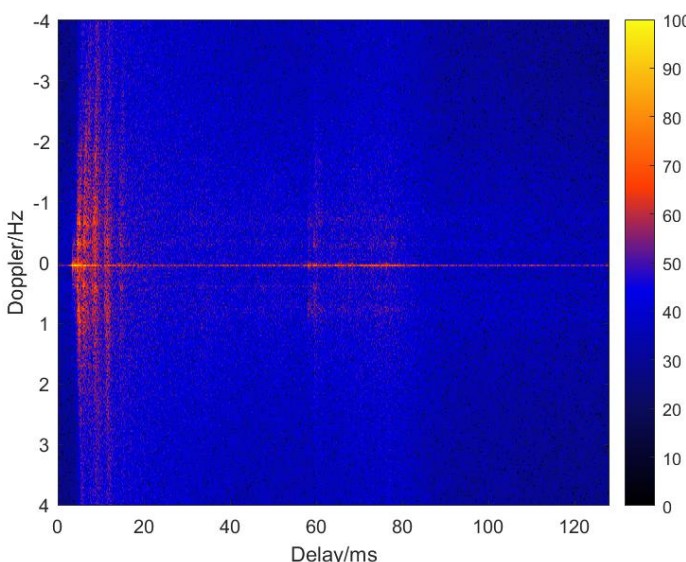

**Figure 8.** CIR in DD domain under NOF experiment.

As shown in Figure 9, the BER performance of the proposed SC-CNN-BiLSTM-based signal detection method was compared with other methods. Similar to the results in the simulation channel, the proposed SC-CNN-BiLSTM-based signal detection showed the best performance under the NOF UWA experimental channel. The proposed method had 5 dB, 2.5 dB and 2 dB gains at a BER of $1 \times 10^{-3}$ compared with MP-, FC-DNN- and 2D-CNN-based signal detection methods.

Compared with the 2D-CNN, SC-CNN-BiLSTM enhanced the performance by employing skip connections for data reuse and BiLSTM in an RNN for time series data processing. Specifically, in SC-CNN-BiLSTM, skip connections with the CNN can provide more efficient information through the interactions of the convolutional and deconvolutional layers. After the SC-CNN extracts the features of the input vector, LSTM can further focus on effective information in the data sequences to mitigate ICI and ISI. A single CNN can only extract local features and cannot process time series data as efficiently as an RNN.

Compared with DNN-based signal detection, our proposed method utilized a CNN and BiLSTM cascaded network for better data fitting than a single neural network. There are non-convex optimization and gradient disappearance problems in the FC-DNN, which limit its robustness.

Compared with nonlinear MP detection, SC-CNN-BiLSTM can converge to the optimum, whereas MP may get trapped in local optimum and have high complexity during iteration. Compared with linear-based methods, such as LMMSE and ZF, the SC-CNN-BiLSTM signal detection method can use nonlinear operations in the neural network to better fit data in the DD domain.

The CIR of the KAU1 channel in the time and DD domains are shown in Figures 10 and 11, respectively. In both the time and DD domains, the CIR structure was more complex than that for NOF. The channel variations were also more obvious than those for NOF. In the DD domain, it can be seen that the maximum Doppler shift of KAU1 was larger than that for NOF, which was about 4 Hz. Note that in these two sea experiments, although the transmitter and receiver were deployed in fixed locations, the Doppler shift was still evident. The Doppler shift in practical UWA channels is severe and complex, and it is caused by the multiple unique characteristics of the UWA environment. For example, the movement of seawater can cause the transceiver to move.

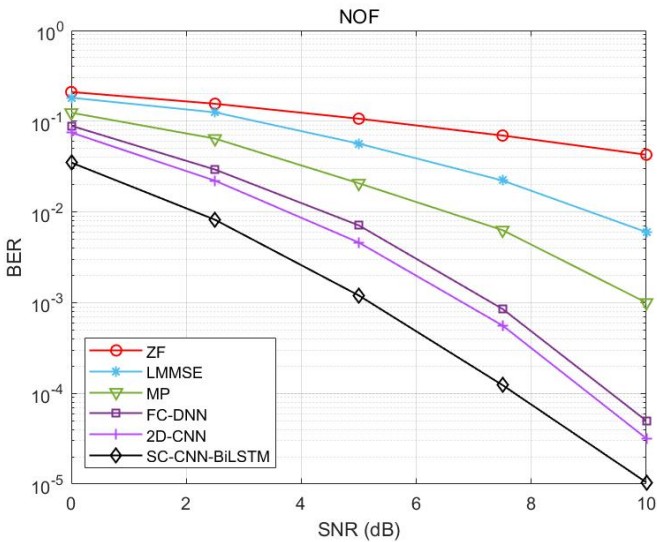

**Figure 9.** BER performance comparison for NOF experiment.

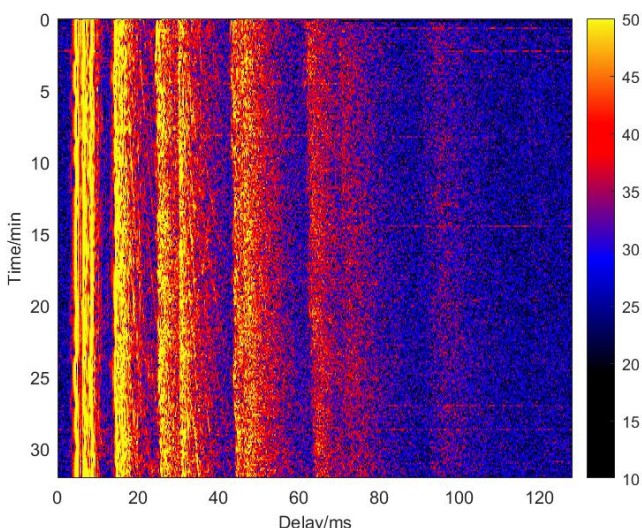

**Figure 10.** CIR in time domain under KAU1 experiment.

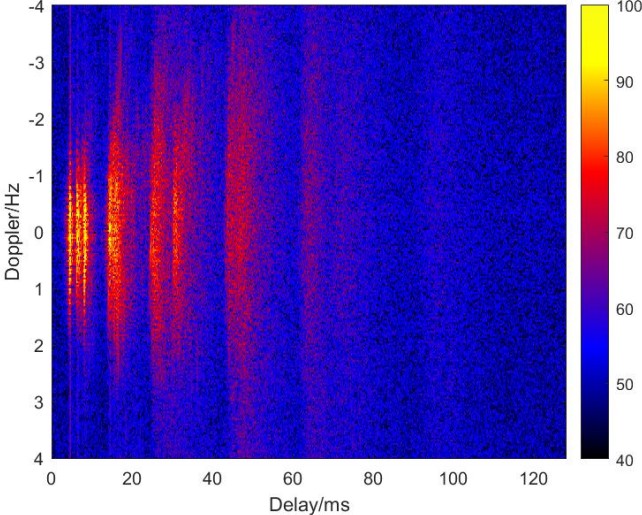

**Figure 11.** CIR in DD domain under KAU1 experiment.

In Figure 12, the BER performance of multiple signal detection methods is similar to that of the NOF channel. In the KAU1 experimental channel, SC-CNN-BiLSTM-based signal detection could achieve 4.5 dB, 2.5 dB and 1.5 dB SNR gains at a BER of $1.5 \times 10^{-3}$ compared with the MP, FC-DNN and 2D-CNN methods, respectively. Owing to the specific design of the neural network, our proposed method outperformed both DL-based signal detection and conventional linear or nonlinear signal detection methods.

When comparing the BER performance of SC-CNN-BiLSTM in NOF (Figure 9) and KAU1 (Figure 12), the BER performance at NOF was better than that at KAU1. The multipath structure and Doppler shift of KAU1 were more severe than those at NOF, which would degrade the BER performance. In the two experimental channels, SC-CNN-BiLSTM-based OTFS detection outperformed all the other signal detection methods.

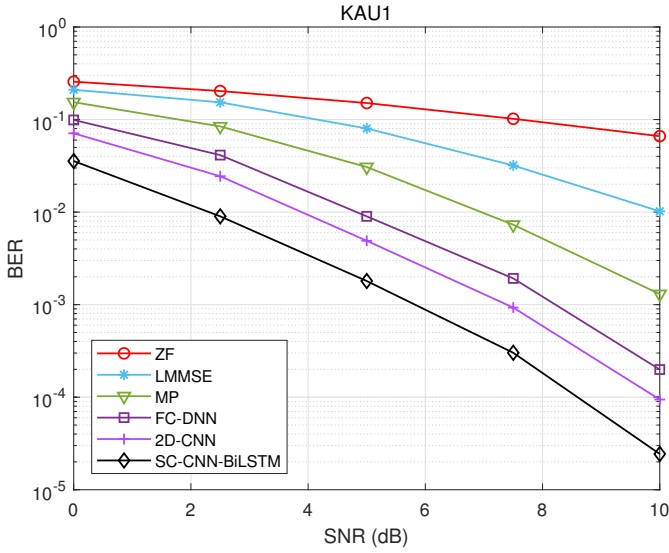

**Figure 12.** BER performance comparison for KAU1 experiment.

### 4.4. Robustness Analysis with UWA Non-Ideal Channel Estimation

In the actual UWA communication system, many uncertainties will make the estimated CSI non-ideal. It can be seen from the literature [35] that when the system obtains the CSI with errors, the performance of the system will degrade. In this subsection, the impact of non-ideal channel estimation on the proposed model will be analyzed. Assume that the channel estimation errors follow a Gaussian distribution with a zero mean and a certain variance.

The KAU1 experimental channel dataset was used to evaluate the multiple signal detection methods with channel estimation error. In the simulation, the variance of the channel estimation error was set to 0.1.

As shown in Figure 13, with the channel estimation error, the BER performances of conventional signal detection methods obviously became worse. In the experimental channel, the BER of ZF and LMMSE almost increased to the error floor, and the BER of MP increased by up to two orders of magnitude.

The BER performances of DL-based signal detection methods degraded less sharply than those for the conventional methods. As shown in Figure 13, compared with the two DL-based signal detection methods, SC-CNN-BiLSTM-based signal detection could recover data more accurately with the channel estimation error. In the KAU1 experimental channel, at a BER of $2 \times 10^3$, the FC-DNN, 2D-CNN and SC-CNN-BiLSTM methods with channel estimation error had 3 dB, 2 dB and 1.5 dB SNR losses compared with the corresponding signal detection method with ideal channel estimation. The robust performance of SC-CNN-BiLSTM signal detection can be attributed to the proposed cascaded network, where the SC-CNN provides information fusion from the interaction of the current layer and the previous layer for more effective signal feature extraction, and BiLSTM can continuously

store and extract valid information with symbols that are sequentially transmitted from the past to the future.

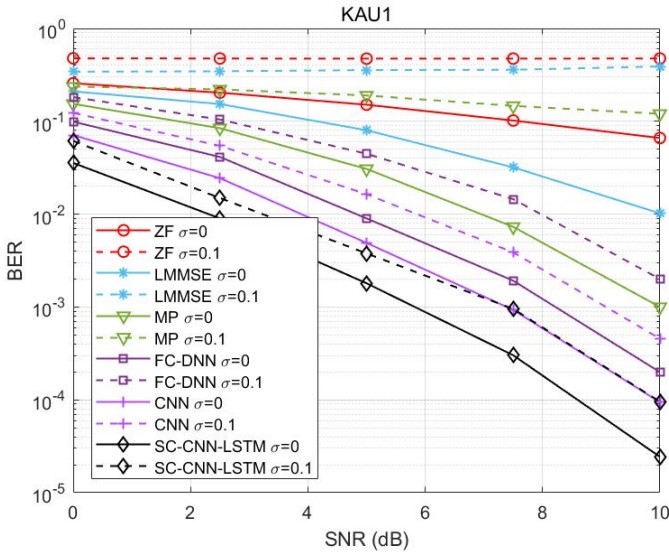

**Figure 13.** BER performance comparison under non-ideal channel estimation for KAU1 experiment.

### 4.5. Computational Complexity Analysis

As shown in Table 5, the computational complexity of the proposed SC-CNN-BiLSTM signal detection and other methods are compared.

**Table 5.** Computational complexity comparison.

| ZF | LMMSE | MP | 2D-CNN | SC-CNN-LSTM |
|:---:|:---:|:---:|:---:|:---:|
| $O((MN)^3)$ | $O((MN)^3)$ | $O(n_{iter}MNn_{ch}\Theta)$ | $O(MN)$ | $O((MN)^2)$ |

The complexity of SC-CNN-BiLSTM can be calculated as the summation of the CNN and LSTM. According to [36], the computational complexity of the CNN can be defined as $O(\sum_{l=1}^{N_l} N_{inc_l} \cdot w^2 \cdot N_{f_l}(NM))$, where $N_l$ is the number of CNN layers used to construct the model, $l$ is the index of a convolutional layer, $N_{f_l}$ is the number of filters (also known as the width) in the $l$th layer, $N_{inc_l}$ is the number of input channels of the $l$th layer, $w^2$ is the kernel size and $NM$ is the OTFS frame size for the network input. As all the other parameters are constant during the training and testing phase, the overall complexity is a linear function of $NM$ expressed as $O(NM)$. Meanwhile, the computational complexity of the BiLSTM can be defined as $O(\sum_{l=1}^{d} n^2 d_c)$, where $d_c$ is the dimension of each cell, $n_{in}$ is the size of the input and $n_{in} = NM$ in our system. The complexity of BiLSTM can be expressed as $O((NM)^2)$. Thus, the SC-CNN-BiLSTM complexity is a linear function expressed as $O((NM)^2)$.

The complexity of the 2D-CNN is $O(NM)$ [32]. The complexity of the MP-based method is $O(n_{iter}MNn_{ch}\Theta)$, where $n_{iter}$ is the number of iterations, $n_{ch}$ is the number of non-zero channel taps and $\Theta$ is the modulation bit size. Thus, the complexity of MP depends on the sparsity level of the channels. The complexity of LMMSE and ZF is $O((MN)^3)$.

From Table 5, we can see that the complexity of SC-CNN-BiLSTM was lower than those of the MP, LMMSE and ZF methods but higher than that of the 2D-CNN. Let us take the consideration of complexity and BER performance together. Both the complexity and BER performance of SC-CNN-BiLSTM outperformed FC-DNN, MP, LMMSE and ZP signal detection. Although the proposed method has higher complexity than the 2D-CNN, it can achieve better BER performance. The complexity of SC-CNN-BiLSTM is a linear function expressed as $O((NM)^2)$, and the complexity of the 2D-CNN is $O(NM)$. Therefore, the proposed SC-CNN-BiLSTM signal detection method has higher complexity than the 2D-CNN. For BER performance, in Figure 6, at a BER of $5 \times 10^{-3}$, the proposed

SC-CNN-BiLSTM could achieve about 1.5 dB improvement compared with the 2D-CNN under a statistic simulation channel model. In Figure 9, at a BER of $1 \times 10^{-3}$, the proposed method could achieve about 2 dB of improvement compared with the 2D-CNN under the NOF channel. In Figure 12, at a BER of $1.5 \times 10^{-3}$, the proposed method could achieve about 1.5 dB of improvement under the KAU1 channel. The proposed method of SC-CNN-BiLSTM had higher complexity than the 2D-CNN, but it could achieve better BER performance in various channel conditions.

## 5. Discussion

### 5.1. Findings and Implications

In this paper, OTFS was investigated for UWA communication with severe Doppler and multi-path effects. Specifically, a DL-based UWA OTFS receiver was designed, considering the characteristics of the UWA OTFS channel in the DD domain. The proposed SC-CNN-BiLSTM-based OTFS signal detection method aims to extract effective signal features with the existing ICI and ISI for reliable communication. The results, findings and implications of this paper are discussed below.

This paper analyzed the OTFS system parameter design for UWA communication. The Doppler shift in UWA communication was much more severe than that of radio communication, as can be seen from Table 1. Even a slight movement of the transceiver would cause an obvious Doppler shift. For effective UWA OTFS communication, this paper discussed the system parameters that are applicable to the UWA channel for the expected performance. The discussed details are in Section 2. The small value of $\Delta f$ made the interference severe, and the small value of $N$ affected the detection accuracy.

This paper proposed a UWA OTFS signal detection method based on the deep neural network. The conventional signal methods usually assume a linear system and a known distribution of noise whose performance will downgrade in the complex actual channel. In the research area of OTFS signal detection, the authors of [11,12] employed the conventional linear signal detection methods ZF and LMMSE. In [13], the Bayesian-based nonlinear algorithm MP was advocated for, and the interference term was assumed to be approximately Gaussian distributed noise. However, in realistic UWA communication, the interference term may not follow the Gaussian distribution. Although nonlinear algorithms can approximate optimal performance through a large number of iterations, their complexity is still much higher than that of LMMSE algorithms. For reliable receiver design, this paper proposed a data-driven, DL-based UWA OTFS receiver. It neither has a linear assumption nor requires prior knowledge of the UWA channel. The semi-experimental results with an experimental sea channel in Figures 9 and 12 showed that our proposed method outperformed the typical LMMSE and MP methods. The results indicate the power of deep learning in non-linear variation wireless communication channels without prior knowledge.

This paper proposed an SC-CNN-BiLSTM-based OTFS signal detection method which takes advantage of both the CNN and RNN for feature extraction and sequential data processing. In the research field of DL-based OTFS signal detection, in [32], the 2D-CNN was used instead of the signal detection, but a single CNN can only extract local features and cannot process time series data efficiently. The authors of [33] conducted a preliminary exploration of FC-DNN-based signal detection for UWA OTFS. However, the FC-DNN has non-convex optimization and gradient vanishing problems, which limits its robustness. In this paper, a DL-based signal detection neural network is proposed for UWA OTFS communication, named SC-CNN-BiLSTM, which cascades the CNN with BiLSTM for UWA OTFS signal detection. Specifically, the proposed SC-CNN-BiLSTM-based OTFS signal detection method employs the CNN to extract signal features and learn the hidden relationship between its input and output. In addition, the SC structure provides information fusion from the interaction of the current layer and previous layers. Furthermore, BiLSTM in the network can continuously memorize and extract effective information with sequentially transmitted symbols from the past to the future. The cascaded neural network structure can extract signal features and effective information to recover distorted received symbols.

As shown in Figures 6, 9 and 12, the proposed method had better BER performance than the FC-DNN- and 2D-CNN-based signal detection methods for UWA OTFS. As shown in Figure 13, the proposed method is most robust when the system has channel estimation errors. The results show that the proposed SC-CNN-BiLSTM signal detection method had better performance in multiple communication channels with different scales of multi-path and Doppler effects. The results indicate that the integration of the CNN and RNN can better extract the signal features of time series data. This thought could be employed in various UWA communication scenarios under complex and varying UWA channels.

### 5.2. Limitations and Future Outlook

Although the proposed method outperformed the existing DL-based OTFS signal detection scheme, there are still limitations to this type of method. The neural network is used as a data-driven based method, which heavily relies on labeled data and empirical parameters, and it is sensitive to data. The generalization ability should be further studied. Furthermore, data-driven deep learning takes neural networks as a black box. The interpretability of this type of model is not as clear as in traditional signal detection methods.

We can take a look at the future outlook of the DL-based receiver of OTFS. First, for a data-driven DL receiver, the joint channel estimation and signal detection can be considered for the whole system's optimization, and then, for a model-driven DL receiver, the expert knowledge can be integrated with deep learning. There are some previous studies considering expert knowledge and deep learning for OFDM systems. For OTFS, researchers have started trying to integrate DL into submodules or train some key parameters by DL. The integration of DL and expert knowledge also seems attractive for future research. Finally, a more powerful structure of deep neural networks could be designed according to the system's features and objectives.

### 6. Conclusions

In this paper, a DL-based signal detection neural network is proposed for UWA OTFS communication, named SC-CNN-BiLSTM. In SC-CNN-BiLSTM, we cascade skip-connected CNN with BiLSTM for UWA OTFS signal detection, which can extract signal features and effective information to recover distorted received symbols. The numerical results show that the proposed method had better BER performance than the ZF, LMMSE, MP, FC-DNN and 2D-CNN signal detection methods in UWA OTFS. This was because the proposed SC-CNN-BiLSTM-based OTFS signal detection method could use the CNN to extract the signal features and learn the hidden relationship between its input and output. In addition, the SC structure provided the information fusion from the interaction of the current layer and previous layers. Furthermore, BiLSTM in the network can continuously memorize and extract effective information with sequentially transmitted symbols from the past to the future, which can mitigate the ICI and ISI. For UWA OTFS, the proposed SC-CNN-BiLSTM signal detection method had better performance and low complexity in multiple communication channels with different scales of multi-path and Doppler effects.

**Author Contributions:** Conceptualization, X.S.; data curation, S.Z.; formal analysis, Y.Z.; funding acquisition, B.W.; investigation, Y.Z.; methodology, S.Z., B.W. and W.B.; project administration, Y.Z. and B.W.; resources, W.B. and X.S.; software, S.Z.; supervision, Y.Z. and B.W.; validation, W.B. and Y.L.; visualization, S.Z.; writing—original draft, S.Z.; writing—review and editing, Y.Z. and Y.L. All authors have read and agreed to the published version of the manuscript.

**Funding:** This research was funded by the National Natural Science Foundation of China under grant no. 61801372, U19B2015, 62001360, 62031021 and 61801371 and the Scientific Research Program of Shaanxi Education Department. under grant No. 22JK0454.

**Institutional Review Board Statement:** Not applicable.

**Informed Consent Statement:** Not applicable.

**Data Availability Statement:** Publicly available datasets were analyzed in this study. These data can be found here: https://www.ffi.no/forskning/prosjekter/watermark, accessed on 30 June 2022.

**Conflicts of Interest:** The authors declare no conflict of interest.

**Abbreviations**

The following abbreviations are used in this manuscript:

| | |
|---|---|
| UWA | underwater acoustic |
| DL | deep learning |
| OTFS | orthogonal time frequency space |
| SC | skip connections |
| OFDM | orthogonal frequency division multiplexing |
| 2D | two-dimensional |
| TF | time frequency |
| ISI | intersymbol interference |
| BER | bit error rate |
| ICI | inter-carrier interference |
| ZF | zero forcing |
| SNR | signal-to-noise ratio |
| LMMSE | linear minimum mean squared error |
| MP | message passing |
| CNN | convolutional neural network |
| LSTM | long short-term memory |
| ISFFT | inverse symplectic finite Fourier transform |
| CIR | channel impulse response |
| SFFT | symplectic finite Fourier transform |
| DeCNN | deconvolutional neural network |
| SVM | support vector machine |
| SGD | stochastic gradient descent |
| Adam | adaptive momentum |
| ReLU | rectified linear unit |

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
