# Peer review of "Deep Learning-Based Signal Detection for Underwater Acoustic OTFS Communication"

_jmse, doi:10.3390/jmse10121920_

Round 1

Reviewer 1 Report

The paper is nicely be written by the authors. 

The methods and results look good.

However, the authors are requested to revise their manuscript by considering the following issues:

1. Line 228  "statistic channel simulation model"- please provide better clarification on this ya.

2. Explain further on Lines 326-327:  Although the proposed method SC-CNN-BiLSTM has 326 higher complexity than 2D-CNN, it can achieve better BER performance- provide better clarification and any supporting facts/results

3. Lines 204-205: The training is performed using the root mean square propagation (RMSprop) optimizer- Explain and provide justification what happen with other optimizers? Most probably, we conduct another experiments? 

4. Provide any coding/open sources for computer implementation using TensorFlow and Keras.

5. Maybe can add some details on parameter settings etc. 

Author Response

We greatly appreciate your review and comments on our submitted manuscript. We have revised our manuscript according to the comments and suggestions. These changes are marked up in the manuscript. We next address your concerns in the order you posed them.

Point 1: Line 228 "statistic channel simulation model"- please provide better clarification on this.

Reply: The "statistic channel simulation model" have been described more in details. We added the number of multi-paths, the statistics of each multi-path, and the calculation method of Doppler, etc. A table is also added to clarify the channel model parameters.

Point 2: Explain further on Lines 326-327: Although the proposed method SC-CNN-BiLSTM has 326 higher complexity than 2D-CNN, it can achieve better BER performance- provide better clarification and any supporting facts/results.

Reply: The complexity of SC-CNN-BiLSTM is a linear function as O((NM)^2), and the complexity of 2D-CNN is O(NM). So the proposed SC-CNN-BiLSTM signal detection method has higher complexity than 2D-CNN. Although the proposed method has higher complexity than 2D-CNN, it can achieve better BER performance. As can be seen from Figure 5, at the BER of 5×10^-3, the proposed SC-CNN-BiLSTM signal detection can achieve about 1.5 dB improvement compared with 2D-CNN method under statistic simulation channel model. In Figure 8, at the BER of 1×10^-3, the proposed SC-CNN-BiLSTM can achieve about 2 dB improvement compared with 2D-CNN under NOF channel. In Figure 11, at the BER of 1.5×10^-3, the proposed method can achieve about 1.5 dB more improvement than 2D-CNN under KAU1 channel. So the conclusion is that the proposed method SC-CNN-BiLSTM has higher complexity than 2D-CNN, but it can achieve better BER performance in various channel conditions.

Point 3: Lines 204-205: The training is performed using the root mean square propagation (RMSprop) optimizer- Explain and provide justification what happen with other optimizers? Most probably, conduct another experiment?

Reply: We tested other optimizers, such as SGD and Adam optimizer, and found that their results of the training do not converge well. The convergence of RMSprop optimizer is the best. The specific experiment results figure is added to the paper.

Point 4:  Provide any coding/open sources for computer implementation using TensorFlow and Keras.

Reply: We put the main the network structure code on GitHub. The link is: https://github.com/shumin-Zhang-biteyy/SC-CNN-BiLSTM/tree/main.

Point 5:Maybe can add some details on parameter settings etc.

Reply: Thank you for the advice. More detailed parameter setting of SC-CNN-BiLSTM signal detector is described in the manuscript. The detailed parameters setting of SC-CNN-BiLSTM neural network is updated in Table 2, including the size of the filter, the convolutional kenal, and the stride in the SC-CNN, as well as the size of the layers in the BiLSTM. The parameters are the optimal choices obtained after repeated testing.

Reviewer 2 Report

1-      Explain the importance of using different deep learning models compared to other machine learning algorithms in the introduction section. You can get help from the following article: https://doi.org/10.1016/j.compbiomed.2021.104728

2-      Please compare your results and research with the results of other authors and underline what is the scientific novelty in this work.

3-      The interpretation of results should be well written in the paper. The results need further technical discussions.

4-      The Limitations of the proposed study need to be discussed before conclusion.

5-      Author should add separate section regarding future outlook and specific comment point wise based on their study.

Author Response

We greatly appreciate your valuable comments on our submitted manuscript. We have revised our manuscript according to the comments and suggestions. We next address your concerns and suggestions in the order you posed them.

Point 1: Explain the importance of using different deep learning models compared to other machine learning algorithms in the introduction section. You can get help from the following article: https://doi.org/10.1016/j.compbiomed.2021.104728

Reply: Thank you for the advice. We've added the importance of using different deep learning models compared to other machine learning algorithms in the introduction, and added the recommended articles.

Point 2: Please compare your results and research with the results of other authors and underline what is the scientific novelty in this work.

Reply: Thank you for the advice. In Section 4, we described the proposed method and compared methods of other authors in detail. The result has also described in more details. The scientific novelty of this work is underline in introduction, experiment results analysis and discussion.

Point 3: The interpretation of results should be well written in the paper. The results need further technical discussions.
Reply: Thank you for the advice. We have added more detailed explanations of the simulation results, as well as further technical discussion of these conclusions. The performance comparison of all neural networks and traditional methods are analyzed in more details.

Point 4: The Limitations of the proposed study need to be discussed before conclusion.

Reply: Done. We added the limitations of the proposed study in section 5.2. The limitations are: The neural network is used in this paper as data-driven based method, which heavily relies on labeled data and empirical parameters, and is very sensitive to data. And data-driven based deep learning takes neural networks as a black box. The interpretability of this type of model is not so clear as traditional signal detection methods.

Point 5: Author should add separate section regarding future outlook and specific comment point wise based on their study.

Reply: Done. We added future outlook and specific point in section 5.2. Our main points are: (1) In the future we will focus on deep neural network based UWA OTFS communication systems with joint channel estimation and signal detection. (2) The deep neural network is used in this paper as data-driven end-to-end receiver. There are also studies in which neural networks replace a submodule in OTFS systems or optimize certain parameter of traditional algorithms. The combination of deep learning and expert knowledge can be further studied in the future. (3) Design of neural network structure.

Reviewer 3 Report

1. What are the contributions of the proposed work?

2. How this work is different from others?

3. Please give more description about your future work as well as the limitations of the proposed method.

4. Paper is required to be proofread and checked thoroughly.

Author Response

We greatly appreciate your review and comments on our submitted manuscript. We have revised our manuscript according to the comments and suggestions. And we also checked the writing of the article, corrected grammatical errors, and revised the expression. These changes are marked up in the manuscript. The responses to the comments are listed below point-by-point.

Point 1: What are the contributions of the proposed work?

Reply: Thank you for the question. Our contributions are as follows:

  • The severe Doppler effect of underwater acoustic (UWA) channel makes signal detection challenging. Thus, in this paper, we propose a UWA OTFS signal detection method based on neural network that neither has linear assumption, nor requires prior knowledge of UWA channel compared to other traditional methods. To the best of our knowledge, this is the first DL-based signal detector proposed for UWA OTFS communication.
  • The SC-CNN-BiLSTM network is designed for OTFS signal detection in the complex UWA channel, which takes the advantage of both CNN and RNN for feature extraction and sequential data processing. In this paper, we cascade CNN with BiLSTM for UWA OTFS signal detection to get lower BER than existing DL OTFS receivers. The SC-CNN in the network can extract signal features and learn the hidden relationship between its input and output. BiLSTM in the network can continuously memorize and extract effective information with sequentially transmitted symbols from the past to future.

We strengthen the contribution in the manuscript.

Point 2: How this work is different from others?

Reply: Thank you for the question. The differences include the following two points:

  • In the existing DL-based signal detection of OTFS, a single FC-DNN or CNN is usually used to implement OTFS signal detection. This paper proposed a OTFS signal detection scheme based on joint CNN and RNN. The cascaded neural network structure can extract signal features and effective information to recover distorted received symbols. Especially, RNN has strong ability to capture the correlations of times series data. For the signal detection of sequential data with interference, it can mitigate the interference from neighbor symbols.
  • We design the specific neural network structure of SC-CNN-BiLSTM for OTFS signal detection. We add symmetrical skip connections to transfer learned features mapping from the previous layer to current layer in SC-CNN layer. And we integrated the BiLSTM in RNN to learn more comprehensive intrinsic correlation of input series signal by learning from past to future and from future to past.

We strengthen the differences in the manuscript.

Point 3: Please give more description about your future work as well as the limitations of the proposed method.

Reply: In the future, we will focus on deep learning based UWA OTFS communication systems with joint channel estimation and signal detection. The limitation is that the neural network requires large amount of training data and sensitive to the dataset. The limitations and future work are discussed in details in the Discussion Section.

Point4: Paper is required to be proofread and checked thoroughly.

Reply: We make our effort to check the manuscript sentence by sentence, correct grammatical mistakes, and revise the expressions. We hope our revision makes the paper more professional and understandable.

Round 2

Reviewer 1 Report

Revised accordingly.

Just improve the sentences.

Author Response

Thank you so much for the advice. The writing has been checked and improved.